# Hepatectomy or/with Metastatectomy for Recurrent Intrahepatic Cholangiocarcinoma: Of Promise for Selected Patients

**DOI:** 10.3390/jpm12040540

**Published:** 2022-03-29

**Authors:** Chun-Yi Tsai, Shang-Yu Wang, Kun-Ming Chan, Wei-Chen Lee, Tse-Ching Chen, Ta-Sen Yeh, Yi-Yin Jan, Chun-Nan Yeh

**Affiliations:** 1Department of General Surgery, Chang Gung Memorial Hospital, Linkou Branch, Chang Gung University, Taoyuan 333, Taiwan; andreas3048@gmail.com (C.-Y.T.); shangyuwang@gmail.com (S.-Y.W.); chankunming@cgmh.org.tw (K.-M.C.); weichen@cgmh.org.tw (W.-C.L.); tsy471023@cgmh.org.tw (T.-S.Y.); 2Liver Research Center, Chang Gung Memorial Hospital, Linkou Branch, Chang Gung University, Taoyuan 333, Taiwan; 3Department of Pathology, Chang Gung Memorial Hospital, Linkou Branch, Chang Gung University, Taoyuan 333, Taiwan; ctc323@cgmh.org.tw

**Keywords:** intrahepatic cholangiocarcinoma, recurrent intrahepatic cholangiocarcinoma, hepatectomy, metastatectomy, overall survival

## Abstract

**Introduction:** Intrahepatic cholangiocarcinoma (ICC) has devastating outcomes owing to its advanced stage at diagnosis and high recurrence after hepatectomy. There is no preferred treatment for recurrent ICC. We retrospectively reviewed our patients who underwent repeated operations for recurrent ICCs based on their different indications to appraise the outcomes. **Methods:** In all, 160 out of 216 patients with ICC (71.4%) experienced recurrence after curative resection from 1977 to 2014. The patterns of recurrence were categorized according to the locations and numbers of recurrent tumors. **Results:** Patients with merely intrahepatic recurrence (*n* = 38) had superior overall survival (OS) compared with those with beyond intrahepatic recurrence (*p* < 0.0001). Twenty-seven out of 160 patients (16.8%) underwent repeat hepatectomy or/with metastatectomy for recurrence and had superior OS when compared to the remaining 133 patients who received nonoperative treatment/palliation (85.6 months versus 20.9 months, *p* < 0.001). Furthermore, patients suitable for repeat hepatectomy in the intrahepatic recurrent group (*n* = 12) had superior post-recurrence overall survival (PROS) than the remaining 26 patients receiving nonoperative treatment (61.6 months versus 14.7 months, *p* < 0.05). **Conclusion:** Liver is the most commonly involved site of recurrent ICC. However, merely intrahepatic recurrence may have a favorable prognosis compared to recurrence involving other sites. Aggressive hepatectomy may provide a survival benefit in selected patients.

## 1. Introduction

Intrahepatic cholangiocarcinoma (ICC) is the second most common primary hepatic malignancy, although its incidence is relatively rare when compared with other gastrointestinal malignant tumors [1,2]. Importantly, it has been more prevalent in Asia, though incidence in Western countries might have been underestimated in recent decades [3]. Curative hepatectomy remains the best treatment modality to achieve potentially long-term survival for ICC; however, the results are usually devastating, stemming from the high recurrence rate and advanced stage at diagnosis [4,5,6,7]. To date, either a standard or a universal guideline or regimen for resected biliary cancer has not yet been proposed, although there are several ongoing trials [8,9]. The optimal method to treat recurrent ICC remains undetermined, although various single or combined treatment modalities have been proposed with acceptable outcomes, including repeat resection, radiofrequency ablation (RFA), concurrent chemoradiation, transarterial chemoembolization (TACE), and yttrium-90 (Y-90) radioembolization [10,11,12]. Most of the results of these locoregional therapies were based on small cohorts of unresectable biliary tract cancer (sometimes combined with gallbladder cancer), while some were based on unresectable ICCs. The results showed that locoregional therapies could be safe and effective; however, there is still no consensus on treatment for recurrent ICC due to scant case numbers. During our practice, we observed patients suffering from intrahepatic recurrent ICCs that seemed to be suitable for a second hepatectomy. Although repeat hepatectomy is a part of the liver-directed treatment, patients suitable for repeat hepatectomy for recurrent ICC are extremely rare for the following reasons: insufficient hepatic remnant after previous surgeries, disseminated or multifocal intrahepatic recurrence, or medical unfitness. However, hepatectomy might still be a rational effective treatment for patients with scan images showing solitary intrahepatic recurrent tumors. In order to explore the predictive factors for recurrence and to clarify the potential benefit of repeat hepatectomy or/with metastatectomy, we retrospectively reviewed our patients suffering from recurrent ICC after curative resection who received an operation alongside other liver-directed treatment.

## 2. Materials and Methods

Between January 1977 and December 2014, 452 patients underwent curative resection with different extent of hepatectomy for ICC in the Department of General Surgery, Chang Gung Memorial Hospital, Linkou branch, Taiwan. The diagnosis of ICC was confirmed by pathologic examination after resection. This study was approved by the Institutional Review Board of Chang Gung Memorial Hospital (Approval number: 201701127B0). The study enrolled 216 patients who underwent R0 resection (58.4%) at their first hepatic resection for ICC. Patients with R1/R2 resections, distant seeding tumors confirmed during laparotomy, and those who suffered from surgical mortality were excluded (Figure 1). According to our previous report, patients with positive margins after hepatectomy for primary ICC were also excluded to minimize the negative impact on survival [13]. The clinicopathological features of these 216 patients were collected and analyzed. The last date of follow up was 31 December 2016. We defined recurrence based on one of the following features: (1) pathologically proven cholangiocarcinoma at second hepatectomy/tumor excision or percutaneous biopsy, or (2) evidently new lesions on either computed tomography (CT), magnetic resonance imaging (MRI), or positron emission tomography (PET). The sites of recurrence, which could be multifocal, and the treatment after recurrence of each patient were reviewed and analyzed. Recurrence-free survival (RFS) was defined as the duration between curative hepatectomy and the first recognized recurrence. Post-recurrence overall survival (PROS) was defined as survival after the first recognized recurrence until the date of death related to ICC. Overall survival (OS) was defined as the survival after the diagnosis of ICC until the date of death related to ICC. All data are presented as the percentage of patients or mean with standard deviation or median with 95% confidence of interval (CI). Numerical data were compared by independent two-sample Student t-tests. Nominal data were compared using the Pearson chi-square test, Fisher exact test or multiple forward stepwise logistic regression test, as appropriate. The survival curves were produced by the Kaplan–Meier method and the survival difference was compared by the log rank test. We performed all statistical analyses using IBM SPSS Statistics for Windows (ver. 20.0; IBM, Chicago, IL, USA). *p* < 0.05 was considered statistically significant.

## 3. Results

### 3.1. Clinical Outcome of 216 Patients with ICC Who Underwent Curative Hepatectomy

Table 1 summarizes the clinicopathological features of the 216 patients with ICC undergoing curative hepatectomy. The mass-forming (MF) type (123/216; 56.9%) was the most common gross pathological type, followed by the intraductal papillary (42/216, 19.4%) and periductal infiltrating types (30/216, 13.9%). The median follow-up time for all patients was 26.9 months (range: 1.7~268.0 months). For 216 patients with ICC undergoing curative hepatectomy, the RFS rates were 57.5%, 33.0%, and 26.1% at 1 year, 3 years, and 5 years, respectively (median RFS: 15.6 months with 95% CI from 11.0 to 20.2 months). The OS rates were 84.2%, 45.7%, and 33.9% at 1 year, 3 years, and 5 years, respectively (median OS: 37.2 months with 95% CI from 26.0 to 39.4 months). 

### 3.2. Distribution of Recurrence, Predictive Factors, and the Relationship between the Recurrent Sites and the Prognosis

A total of 160 patients out of the cohort (74.1%) experienced recurrences involving different sites during follow-up. Table 2 shows the predictive factors for recurrence of 160 ICC patients who underwent curative hepatectomy, irrespective of the recurrence site. Large tumor size (tumor size larger than 5 cm) with positive vascular invasion was independently associated with higher risk of recurrence, as demonstrated by a multivariate logistic regression model. According to the locations and numbers of the recurrent tumors, the pattern of recurrence of the 160 patients could be stratified into three partially overlapping circles, as intrahepatic involvement, locoregional involvement, and distant metastasis (Figure 2), since some of the patients had multifocal recurrence. Locoregional recurrence was defined as tumors located around the hepatoduodenal ligament, the lesser curvature side of the stomach, the paraduodenal region, and the peripancreatic area. Tumors located beyond these areas were designated distant metastases. The liver was involved in 47% (75 of 160) of the recurrences, either isolated or concomitant with other extrahepatic recurrences. Table 3 shows the related predictive factors among different recurrent sites. For patients with involvement of liver as the recurrent site, gross pathological morphology and vascular invasion were the two independently predictive factors. For locoregional relapse, larger tumor size was the only independently predictive factor. For patients with distant metastasis, larger tumor size and hepatolithiasis were the two independently predictive factors. Next, the cohort was stratified into three new groups without overlap (Figure 2, different greyscale areas): Group A, recurrent tumors limited to the intrahepatic region (*n* = 38); Group B, locoregional recurrence beyond the liver (*n* = 57); and Group C, distant metastasis (*n* = 65). Group A demonstrated favorable OS compared with groups B and C (18.9 months versus 9.3 months versus 3.8 months, *p* < 0.0001; Figure 3). 

### 3.3. Treatment Outcome of Patients with Recurrence Who Underwent Surgery

Regarding the treatment modalities for recurrence, the majority of the patients received best supportive care (36.3%, 58 of 160), followed by palliative chemotherapy (27.5%, 44 of 160), repeated operations (16.9%, 27 of 160), palliative concurrent chemoradiotherapy (8.8%, 14 of 160), and other treatment modalities, including TACE alone, and so on (10.6%, 17 of 160). The surgical procedures for the 27 patients included repeat hepatectomy alone, tumor excision (metastatectomy) alone, or both. Considering the preoperative presumed planning for recurrent tumors, the majority of the indications were limited intrahepatic tumors on preoperative images (*n* = 23), while the remaining four patients underwent surgery for distant metastatic tumors for symptomatic relief. However, 11 of the 23 patients were found to be carrying extrahepatic tumors during explorative operation and underwent concomitant metastatectomy. Appendix A describe the details of the 12 patients who underwent isolated second hepatectomy and the 11 patients who underwent both hepatectomy and metastatectomy, respectively. The 27 patients who underwent repeated surgeries had significantly more favorable OS than those who underwent treatment/palliation other than surgery (median OS: 85.6 months versus 20.9 months, *p* < 0.001; Figure 4).

### 3.4. The Outcomes of Surgery for Recurrence among the Selected Groups

Figure 5 summarizes the number of patients who underwent repeat hepatectomy alone in group A (*n* = 12), both hepatectomy and metastatectomy in group B (*n* = 11), and metastatectomy in group C (*n* = 4). The PROS rates of these selected and stratified subgroups were calculated. Regarding the PROS between the patients who underwent repeat hepatectomy alone (12 patients) and those who did not (26 patients) in group A, the former subgroup demonstrated a superior median PROS than the latter subgroup (61.6 months versus 14.7 months, *p* < 0.05; Figure 6). Similarly, the patients in group B who underwent hepatectomy or/with metastatectomy (11 patients) yielded superior median PROS compared to those who did not (*n* = 46) (29.2 months versus 8.2 months, *p* < 0.05; Figure 6).

## 4. Discussion

Cholangiocarcinoma is a malignancy originating from the epithelium of the biliary tract and is classified into intrahepatic, perihilar, and extrahepatic according to the anatomical location of the primary tumor. ICC is the least prevalent among the three subtypes and has different biochemical characteristics and behaviors than extrahepatic cholangiocarcinoma [14]. Being known for its advanced stage at diagnosis and its high recurrence rate after hepatectomy, the treatment outcome for ICC remains dismal, leading to limited reports focusing on the surgical treatment for recurrent ICC. The number of patients suitable for surgical intervention for recurrent ICC is scant [4,15,16,17,18,19,20]. In this study, we explored the pattern of recurrence after curative resection for ICC and the treatment outcomes, with several important issues emerging with implications for clinical practice.

First, for recurrence after curative hepatectomy, the liver was the most commonly involved site (47.5%, 76 of 160). Patients with isolated intrahepatic recurrence had a significantly more favorable outcome than patients with recurrence beyond the liver, irrespective of treatment modality. However, our cohort contained only 38 patients who had isolated intrahepatic recurrence (23.8%). The above findings might partially explain the fact that a high recurrence rate for ICC following surgery with unfavorable recurrence patterns (beyond hepatic recurrence) leads to dismal treatment outcomes.

Second, similar to a multicenter study described by Spolverato [21], surgery provided a significant survival benefit for selected patients with isolated-intrahepatic recurrence and local-regional recurrence. Repeat hepatectomy has been advocated as the choice of treatment for recurrent hepatocellular carcinoma or liver metastasis from colon adenocarcinoma [22]. This is supported by our report, which demonstrated that, for highly selected patients with isolated intrahepatic recurrent ICC, receiving hepatectomy resulted in a significantly longer PROS than for those who did not, although the possibility of selection bias cannot be excluded. Figure 7 illustrates that there was no difference between the PROS values of the nonoperative subgroups in group A and group B/C (*n* = 26 and *n* = 107, respectively, Figure 3). Although we demonstrated that limited intrahepatic recurrence represented the best survival chances of all, when a second operation was not feasible, the prognosis was as dismal as for those with extrahepatic recurrence/metastasis. Berry-picking surgery was never a standard treatment for recurrent ICC. Several studies composed of combined modalities for unresectable ICCs yielded acceptable results [23,24,25]. The overall weighted median survival was 15 months based on a systematic review focusing on Y-90 therapy for unresectable ICCs. Nevertheless, these reports were mostly from 2010 and did not include randomized control trials, mainly due to the rarity of cases. Hepatic arterial infusion (HAI) chemotherapy is another choice of liver-directed therapy for unresectable ICCs [26]. Compared to locoregional therapy other than surgery, the median tumor response rate reached over 50%. However, the adverse effect was the highest. Based on the above findings, repeat hepatectomy for selected patients based on the concept of liver-directed therapy remains feasible since there is no superior treatment for recurrent ICCs.

Lastly, looking into the detailed clinicopathological features of the primary ICC of the twelve patients suitable for repeated hepatectomy, eleven of them presented with MF type ICC, while only one had mixed-type ICC by morphological classification (Appendix A). The above observation supported our previous report, and gross pathological classification of ICC determined the efficacy of hepatectomy [27]. Spolverato’s report also demonstrated a similar phenomenon [21]. Regarding recurrent ICC in which the primary tumor is morphological MF type, there have been no previous reports. This observational result warrants further study to clarify the relationship between the gross morphology and the feasibility of repeat operation for recurrent tumors.

Postoperative morbidity and mortality were the major concerns in this specific scenario. In this study, 27 patients underwent a second operation, only one patient developed postoperative bile leakage, and one patient had a laparotomy wound infection (rate of surgical complications: 7.4%). There was no surgical mortality after the second operation. Although patients suitable for more than two surgeries are extremely rare, repeat hepatectomy or/with metastatectomy might be feasible in selected patients by experienced surgeons in view of surgical complications.

Although we report a positive impact of repeat hepatectomy or/with metastatectomy on patients with recurrent ICC, selective and recall bias cannot always be prevented in a retrospective study. The first limitation of the current study is that the study period spanned over 30 years, and the approach to treating recurrent ICC has evolved from decade to decade. We sought to simplify the scenario by selecting patients who received repeat hepatectomy or metastatectomy. Treatment before the operations warrants further analysis. Second, since this was not a double-blinded study, biases owing to patients’ and surgeons’ attitudes were likely to arise, which might have affect the decision to perform surgery for recurrent tumors. To provide more solid evidence of the impact of repeat hepatectomy on recurrent ICC, a prospective, randomized trial is essential.

## 5. Conclusions

In summary, the recurrence rate of ICC after curative resection was high, and aggressive tumor behavior resulted in a high risk of recurrence. The liver was the most commonly involved site of recurrence, whether solitary or combined with other extrahepatic sites, and the pattern of recurrence determined its prognosis. Patients with merely intrahepatic recurrence had a superior PROS compared to the other patterns, while aggressive hepatectomy may provide a survival benefit. In selected patients, undergoing concomitant metastatectomy was associated with a superior PROS compared to patients who were not suitable to undergo concomitant metastatectomy.

## Figures and Tables

**Figure 1 jpm-12-00540-f001:**
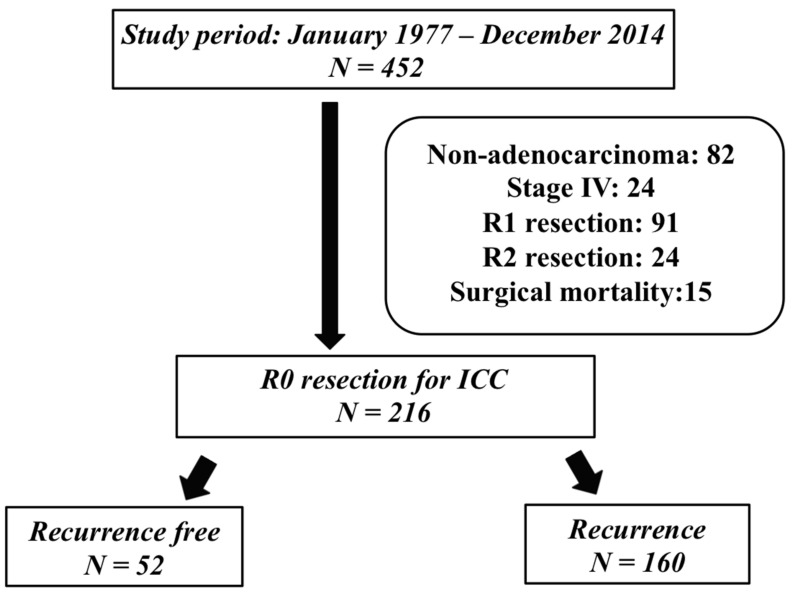
The algorithm and flowchart of patient selection.

**Figure 2 jpm-12-00540-f002:**
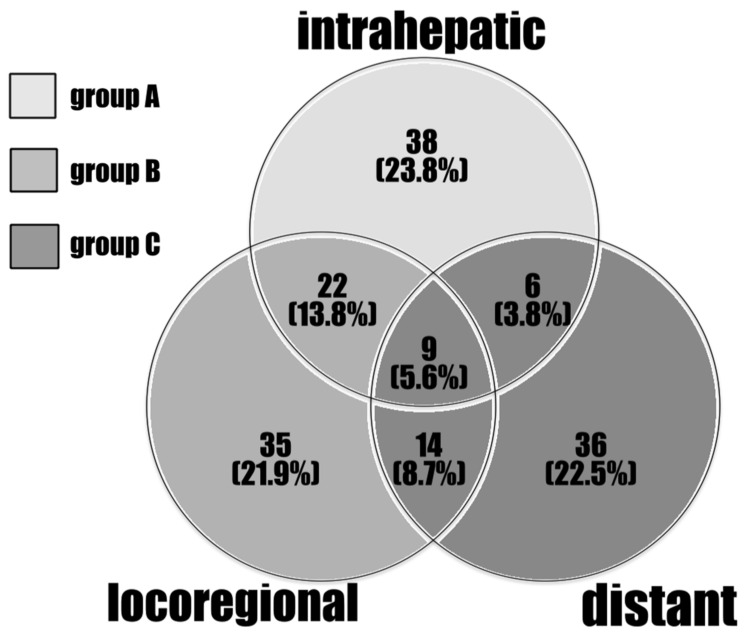
The distribution of recurrent tumors based on their locations and numbers.

**Figure 3 jpm-12-00540-f003:**
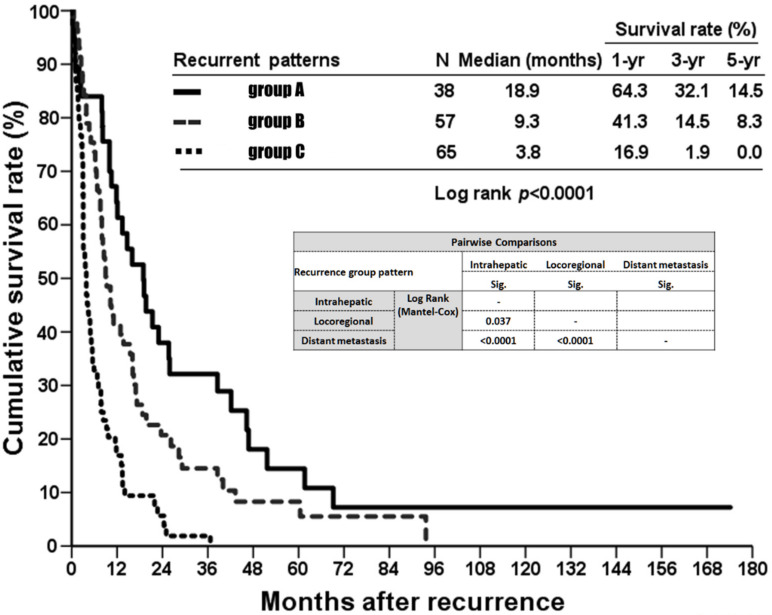
The comparison of overall survival (OS) among the three different recurrent patterns, represented by groups A, B, and C.

**Figure 4 jpm-12-00540-f004:**
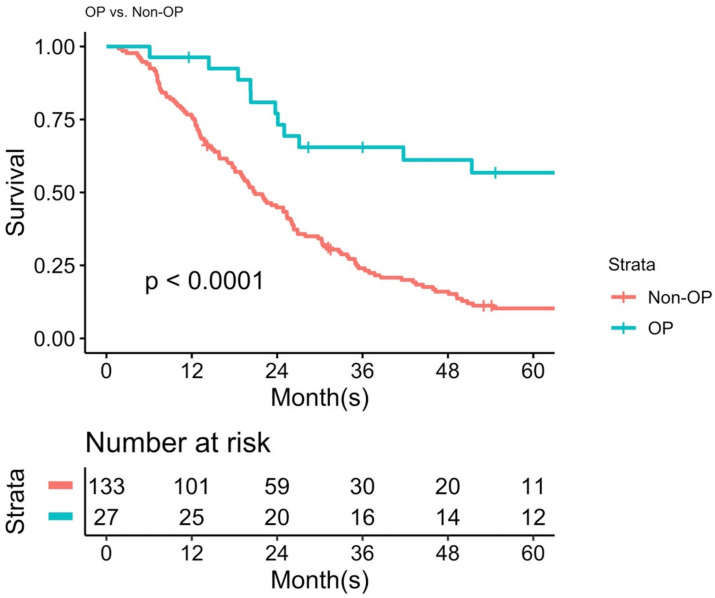
The difference in post-recurrence overall survival (PROS) between the patients who underwent repeat operation and those who did not.

**Figure 5 jpm-12-00540-f005:**
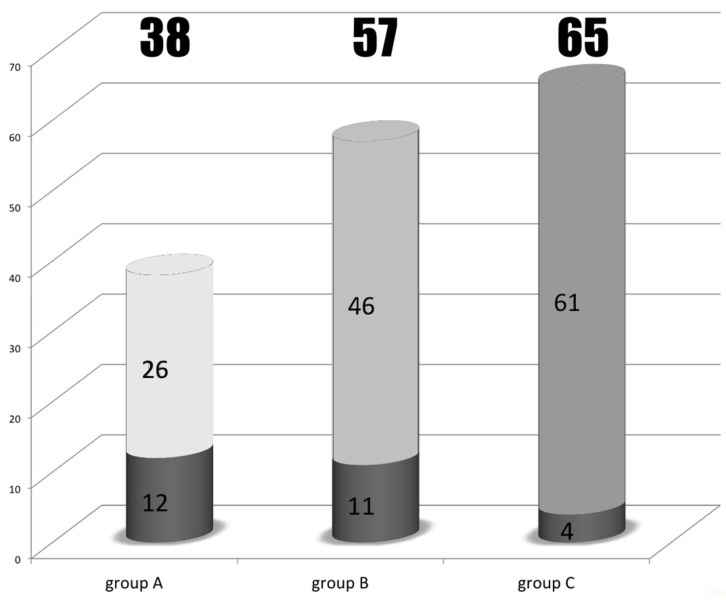
The individual number of patients who underwent a second operation within each group.

**Figure 6 jpm-12-00540-f006:**
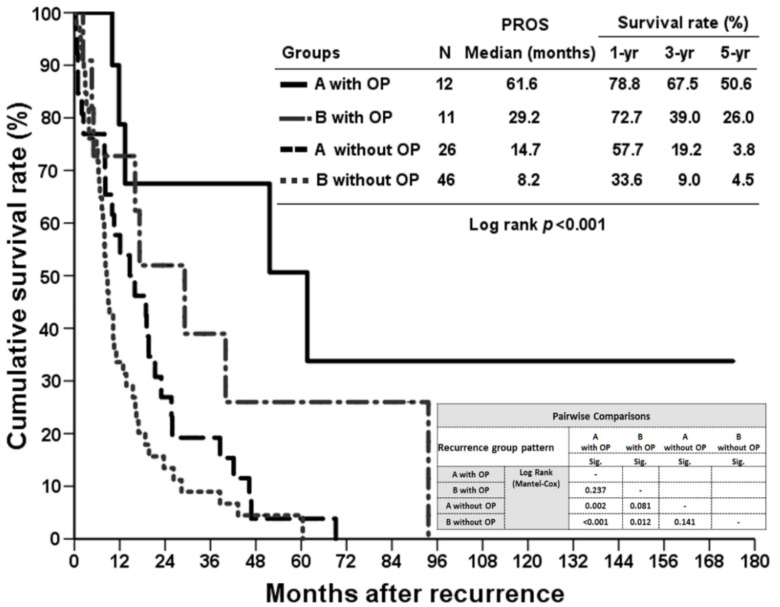
The post-recurrence overall survival (PROS) according to the types of surgery applied within groups A and B. The first line indicates that the patients with isolated intrahepatic recurrence underwent repeat hepatectomy alone in group A (*n* = 12), compared to the third line, which represents the nonoperative patients (*n* = 26) within group A. The second line indicates the patients with locoregional recurrence who underwent hepatectomy or/with metastatectomy in group B (*n* = 11), and the fourth line indicates the nonoperative patients within group B (*n* = 46).

**Figure 7 jpm-12-00540-f007:**
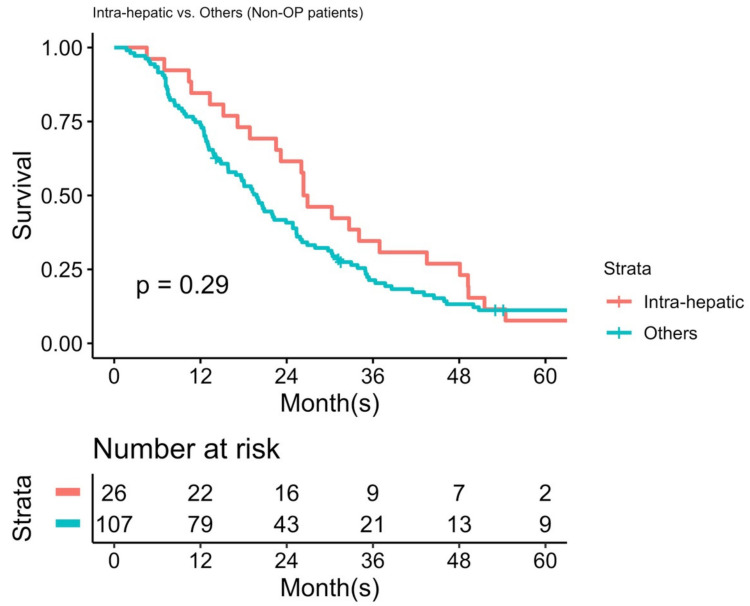
The difference in post-recurrence overall survival (PROS) between the nonoperative patients within group A and the nonoperative patients within groups B and C.

**Table 1 jpm-12-00540-t001:** Clinicopathological characteristics of 216 patients with intrahepatic cholangiocarcinoma achieving curative hepatectomy.

Clinicopathological Characteristics of 216 Patients with Intrahepatic Cholangiocarcinoma Achieving Curative Hepatectomy		
		Mean ± SD or *n* (%)
Age at diagnosis (years)		60.5 ± 11.6
CEA (ng/mL)		19.9 ± 45.4
CA 19-9 (U/mL)		2130.9 ± 9603.7
Associated IHD stones	yes	51 (23.2)
	no	165 (76.8)
Mucobilia	yes	23 (10.6)
	no	193 (89.4)
Hepatitis B		48 (22.2)
Hepatitis C		19 (8.8)
Extent of hepatectomy	Partial	11 (5.1)
	Segmental	19 (8.8)
	Multi-segmental	78 (36.1)
	Left hepatectomy	73 (33.8)
	Right hepatectomy	24 (11.1)
	Extended hepatectomy	11 (5.1)
Tumor size (cm)		5.4 ± 2.6
T stage	T1	108 (50.0)
	T2	23 (10.6)
	T3	34 (15.7)
	T4	51 (23.6)
N stage	N0	180 (83.3)
	N1	36 (16.7)
AJCC 7th TNM stage	I	103 (47.7)
	II	18 (8.3)
	III	24 (11.1)
	IV	71 (32.9)
Histopathology of tumor	Well differentiated	30 (13.9)
	Moderate differentiation	90 (41.7)
	Poorly differentiated	61 (28.2)
	Others	35 (16.2)
Gross morphology	Intraductal papillary	42 (19.4)
	Mass forming	123 (56.9)
	Periductal infiltration	30 (13.9)
	Mixed	21 (9.7)

CEA, carcinoembryonic antigen; CA 19-9, carbohydrate antigen 19-9.

**Table 2 jpm-12-00540-t002:** Univariate and multivariate logistic regression model to predict risk factors for any site recurrence.

	Any Site Recurrence	Recurrence			Multivariate		
Factors		No (*n* = 56)	Yes (*n* = 160)	*p* Value	Odds Ratio	95% CI	*p* Value
Age				0.054			
≤65 (*n* = 135)		29 (21.5)	106 (78.5)		-		
>65 (*n* = 81)		27 (33.3)	54 (66.7)				
Gender				0.604			
Male (*n* = 99)		24 (24.2)	75 (75.8)		-		
Female (*n* = 117)		32 (27.4)	85 (72.6)				
Liver cirrhosis				0.771	-		
No (*n* = 195)		50 (25.6)	145 (74.4)				
Yes (*n* = 21)		6 (28.6)	15 (71.4)				
IHD stones				0.655	-		
No (*n* = 165)		44 (26.7)	121 (73.3)				
Yes (*n* = 51)		12 (23.5)	39 (76.5)				
Gross morphology				0.004			
IG (*n* = 42)		18 (42.9)	24 (57.1)		1		
MF (*n* = 30)		33 (26.8)	90 (73.2)		1.61	0.68–3.85	0.281
Mix (*n* = 123)		2 (9.5)	19 (90.5)		2.51	0.35–18.11	0.361
PI (*n* = 21)		3 (10.0)	27 (90.0)		2.75	0.51–14.69	0.237
Histologic differentiation				0.048			
Well to moderately differentiated (*n* = 151)		45 (29.8)	106 (70.2)		1		
Poorly to undifferentiated (*n* = 65)		11 (16.9)	54 (83.1)		1.55	0.66–3.61	0.314
Primary tumor size (cm)				0.004			
≤5 (*n* = 96)		33 (34.4)	63 (65.6)		1		
>5 (*n* = 106)		18 (17.0)	88 (83.0)		2.06	1.01–4.22	0.048
T stage				<0.001			
T1-2 (*n* = 131)		45 (34.4)	86 (65.6)		1		
T3-4 (*n* = 85)		11 (12.9)	74 (87.1)		1.44	0.50–4.14	0.502
N stage				0.008			
N0 (*n* = 180)		53 (29.4)	127 (70.6)		1		
N1 (*n* = 36)		3 (8.3)	33 (91.7)		2.37	0.61–9.15	0.211
Vascular invasion				0.001			
No (*n* = 183)		55 (30.1)	128 (69.9)		1		
Yes (*n* = 33)		1 (3.0)	32 (97.0)		9.71	1.25–75.65	0.030
Lymphatic invasion				0.060			
No (*n* = 184)		52 (28.3)	132 (71.7)		-		
Yes (*n* = 32)		4 (12.5)	28 (87.5)				
Perineural invasion				0.010			
No (*n* = 166)		50 (30.1)	116 (69.9)		1		
Yes (*n* = 50)		6 (12.0)	44 (88.0)		1.23	0.42–3.61	0.703

**Table 3 jpm-12-00540-t003:** Multivariate logistic regression model to predict risk factors for different recurrence patterns.

	Recurrence Pattern	Any Site (*n* = 160)			Intrahepatic (*n* = 75)			Locoregional (*n* = 80)			Distant (*n* = 65)		
Variables		OR	95% CI	*p* Value	OR	95% CI	*p* Value	OR	95% CI	*p* Value	OR	95% CI	*p* Value
IHD stones		-						-					
No					1.21	0.50–2.91	0.677				1		
Yes					1						2.64	1.28–5.45	0.009
Gross morphology								-			-		
IG		1			1								
MF		1.61	0.68–3.85	0.281	3.32	1.18–9.32	0.023						
Mix		2.51	0.35–18.11	0.361	7.73	2.02–29.54	0.003						
PI		2.75	0.51–14.69	0.237	1.42	0.40–5.02	0.582						
Differentiation					-			-			-		
Well to moderately		1											
Poorly		1.55	0.66–3.61	0.314									
Primary tumor size					-								
≤5		1						1			1		
>5		2.06	1.01–4.22	0.048				2.09	1.14–3.81	0.017	2.20	1.15–4.23	0.017
T stage					-			-			-		
T1-2		1											
T3-4		1.44	0.50–4.14	0.502									
N stage					-						-		
N0		1						1					
N1		2.37	0.61–9.15	0.211				2.19	0.94–5.10	0.071			
Vascular invasion								-			-		
No		1			1								
Yes		9.71	1.25–75.65	0.030	3.87	1.62–9.24	0.002						
Lymphatic invasion		-									-		
No					1			1					
Yes					0.98	0.38–2.55	0.974	1.26	0.50–3.19	0.632			
Perineural invasion											-		
No		1			1			1					
Yes		1.23	0.42–3.61	0.703	1.34	0.62–2.87	0.459	1.64	0.77–3.52	0.202			

## Data Availability

Not applicable.

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
