# Peer review of "Hepatectomy or/with Metastatectomy for Recurrent Intrahepatic Cholangiocarcinoma: Of Promise for Selected Patients"

_jpm, 2022, doi:10.3390/jpm12040540_

Round 1
Reviewer 1 Report
The manuscript submitted by Chun-Yi-Tsai et al. is an interesting study evaluating the role of hepatectomy or/with metastatectomy in selected patients for the treatment of recurrent intrahepatic cholangiocarcinoma (ICC). The manuscript is of interest given the current challenge to treat cholangiocarcinoma.
The Authors show that patients who had merely intrahepatic recurrence of ICC and underwent surgery had significantly favorable overall survival than the ones who underwent treatment or palliation other than surgery.
Nevertheless the rarity of this disease, this study considered a great number of patients.
A number of issues need to be addressed before coming to final conclusions.
- The Authors should explain the specific features of the patients who underwent second hepatectomy and/or metastatectomy and the ones excluded from surgery. It could be very helpful in the clinical setting.
- It could be interesting to investigate the characteristics of the patients who had merely intrahepatic recurrence and the ones of who had locoregional recurrence/metastasis, in order to try to predict patients’ outcomes and treat them consequently.
- This article states that most of the patients feasible for hepatectomy presented with mass forming (MF) type ICC, so it concludes that “gross pathological classification of ICC determines the efficacy of hepatectomy and provides a higher chance for repeated hepatectomy”. However, it is not clear the frequency of MF type of ICC in the patients who didn’t undergo surgery (is there a prevalence of other types of ICC?): this observation needs to be better discussed and examined more deeply.
- The abstract should be clearer and more organized. Moreover, it is not well defined the main court of the patients and the criteria of selection of patients are not defined accurately. The article provides interesting data, but they could be better commented and organized.
Author Response
Please see the attachment, thank you for your kind review.

Reviewer 2 Report
Although this is a very interesting study regarding the potential significance of hepatectomy or/with metastatectomy for recurrent intrahepatic cholangiocarcinoma, several issues have been raised and major revisions are required.
- The Introduction section is too short. Additional data regarding other therapeutic options should be included. The aim of the survey is not clear.
- In the Methods section, inclusion and exclusion criteria should b emore analysed.
- Results section, should be better presented.
- Figures' quality is acceptable.
- In the Discussion section, parallel evaluation with recent studies' evidence is missing.
- Newly published articles should be included in the References section.
- Grammatical errors should be corrected throughout the Text.
Author Response
Please see the attachment. The revised manuscript was also included.THank you for your kind review and suggestion.

Reviewer 3 Report
Dear Editor, thank you so much for inviting me to revise this manuscript.
This study addresses a current topic.
The manuscript is quite well written and organized. English could be improved.
Figures and tables are comprehensive and clear.
The introduction explains in a clear and coherent manner the background of this study.
We suggest the following modifications:
- Introduction section: although the authors correctly included important papers in this setting, we believe some studies should be cited within the introduction ( PMID: 33571059 ; PMID: 33307876 ), only for a matter of consistency. We think it might be useful to introduce the topic of this interesting study.
- Methods and Statistical Analysis: nothing to add.
- Discussion section: Very interesting and timely discussion. Of note, the authors should expand the Discussion section, including a more personal perspective to reflect on. For example, they could answer the following questions – in order to facilitate the understanding of this complex topic to readers: what potential does this study hold? What are the knowledge gaps and how do researchers tackle them? How do you see this area unfolding in the next 5 years? We think it would be extremely interesting for the readers.
However, we think the authors should be acknowledged for their work. In fact, they correctly addressed an important topic, the methods sound good and their discussion is well balanced.
One additional little flaw: the authors could better explain the limitations of their work, in the last part of the Discussion.
We believe this article is suitable for publication in the journal although major revisions are needed. The main strengths of this paper are that it addresses an interesting and very timely question and provides a clear answer, with some limitations.
We suggest a linguistic revision and the addition of some references for a matter of consistency. Moreover, the authors should better clarify some points.
Author Response
Please see the attachment. The revised manuscript was included also. Thank you for your kind review and previous suggestions.

Round 2
Reviewer 1 Report
The Authors have addressed all issues.
Reviewer 2 Report
As the authors took into consideration all the reviewer's suggestions, the article could be accepted for publication in its current form. Grammatical errors still need correction throughout the Text.
Reviewer 3 Report
Acceptance.